# Changes in Physical Activity and Sedentary Behaviour Due to Enforced COVID-19-Related Lockdown and Movement Restrictions: A Protocol for a Systematic Review and Meta-Analysis

**DOI:** 10.3390/ijerph18105251

**Published:** 2021-05-14

**Authors:** Chris Kite, Lukasz Lagojda, Cain C. T. Clark, Olalekan Uthman, Francesca Denton, Gordon McGregor, Amy E. Harwood, Lou Atkinson, David R. Broom, Ioannis Kyrou, Harpal S. Randeva

**Affiliations:** 1Warwickshire Institute for the Study of Diabetes, Endocrinology and Metabolism (WISDEM), University Hospitals Coventry and Warwickshire NHS Trust, Coventry CV2 2DX, UK; c.kite@chester.ac.uk (C.K.); ad0183@coventry.ac.uk (C.C.T.C.); 2Centre for Active Living, University Centre Shrewsbury, University of Chester, Shrewsbury SY3 8HQ, UK; 3Clinical Evidence Based Information Service (CEBIS), University Hospitals Coventry & Warwickshire NHS Trust, Coventry CV2 2DX, UK; lukasz.lagojda@uhcw.nhs.uk; 4Centre for Intelligent Healthcare, Coventry University, Coventry CV1 5FB, UK; 5Division of Health Sciences, Warwick—Centre for Global Health, Warwick Medical School, University of Warwick, Coventry CV4 7AL, UK; olalekan.uthman@warwick.ac.uk; 6Centre for Sport, Exercise and Life Sciences, Research Institute for Health & Wellbeing, Coventry University, Coventry CV1 5FB, UK; dentonf@uni.coventry.ac.uk (F.D.); ac4378@coventry.ac.uk (G.M.); ad5104@coventry.ac.uk (A.E.H.); ad5173@coventry.ac.uk (D.R.B.); 7Department of Cardiopulmonary Rehabilitation, Centre for Exercise & Health, University Hospitals Coventry & Warwickshire NHS Trust, Coventry CV2 2DX, UK; 8Warwick Clinical Trials Unit, Warwick Medical School, University of Warwick, Coventry CV4 7AL, UK; 9School of Psychology, College of Health and Life Sciences, Aston University, Birmingham B4 7ET, UK; l.atkinson1@aston.ac.uk; 10Aston Medical Research Institute, Aston Medical School, Aston University, Birmingham B4 7ET, UK; 11Warwick Medical School, University of Warwick, Coventry CV4 7AL, UK

**Keywords:** COVID-19, coronavirus, restriction measures, lockdown, physical activity, exercise, sedentary behaviour

## Abstract

Prolonged lockdown/restriction measures due to the COVID-19 pandemic have reportedly impacted opportunities to be physically active for a large proportion of the population in affected countries globally. The exact changes to physical activity and sedentary behaviours due to these measures have not been fully studied. Accordingly, the objective of this PROSPERO-registered systematic review is to evaluate the available evidence on physical activity and sedentary behaviours in the general population during COVID-19-related lockdown/restriction measures, compared to prior to restrictions being in place. Defined searches to identify eligible studies published in English, from November 2019 up to the date of submission, will be conducted using the following databases: CENTRAL, MEDLINE, EMBASE, CINAHL, SPORTDiscus, PSYCinfo, Coronavirus Research Database, Public Health Database, Publicly Available Content Database, SCOPUS, and Google Scholar. The applied inclusion criteria were selected to identify observational studies with no restrictions placed on participants, with outcomes regarding physical activity and/or sedentary behaviour during lockdown/restriction measures, and with comparisons for these outcomes to a time when no such measures were in place. Where appropriate, results from included studies will be pooled and effect estimates will be presented in random effects meta-analyses. To the best of our knowledge, this will be the first systematic review to evaluate one complete year of published data on the impact of COVID-19-related lockdown/restriction measures on physical activity and sedentary behaviour. Thus, this systematic review and meta-analysis will constitute the most up-to-date synthesis of published evidence on any such documented changes, and so will comprehensively inform clinical practitioners, public health agencies, researchers, policymakers and the general public regarding the effects of lockdown/restriction measures on both physical activity and sedentary behaviour.

## 1. Introduction

Since it was first described in December 2019 [1], coronavirus disease 2019 (COVID-19), caused by the highly transmissible and virulent severe acute respiratory syndrome coronavirus-2 (SARS-CoV-2), has evolved into a pandemic which is still ongoing at the time of writing [2,3]. Whilst the majority of patients with COVID-19 are either asymptomatic or exhibit mild symptomatology, COVID-19 can also manifest with severe symptoms, which often require hospitalisation, and potentially intensive care unit support [4,5,6]. Indeed, it is the severity of these cases and the documented fatality of COVID-19 that prompted countries/authorities around the world to implement nationwide lockdown, quarantine, and self-isolation measures [7,8] in order to reduce SARS-CoV-2 transmission rates and protect vulnerable individuals [9,10]. Notably, concerns about SARS-CoV-2 infection [11], and the broader/financial consequences of the pandemic [12], as well as the enforced isolation and reduced social interaction [13] have reportedly also led to an increase in psychological distress in the general population [14].

Physical activity is widely acknowledged as a having a protective effect upon both the physical and mental wellbeing of individuals [15,16], suggesting that it may be a suitable candidate to mitigate the detrimental effects on health caused by the COVID-19-related restriction measures [17]. However, the enforced lockdown/restriction measures, including only being able to exercise outdoors for a limited period per day, have potentially reduced an individual’s opportunities to be physically active, whilst simultaneously providing a situation in which engaging in sedentary behaviours is more commonplace. The latter are also well-known to have significant negative effects on health and quality of life [18,19]. Indeed, in order to promote and maintain health, scientific/health advisors have been vociferous about the need for the general public to engage in regular exercise during the lockdown/restriction measures [20], and this has been reflected in worldwide government recommendations/policies. As such, despite the mandated “stay at home” order during such lockdown periods, in several countries, among the essential reasons for which individuals were allowed to leave their household was to engage in some form of daily exercise [21]. This allowance, and the fact that a significant proportion of non-essential workers were furloughed from their regular jobs [22], may suggest that an often cited barrier [23,24,25] to physical activity participation (i.e., lack of time) was removed during this period. In fact, under these circumstances, the opportunities to be physically active were increased for many individuals affected by the lockdown/restriction measures. In addition, it is also plausible that significant changes to an individual’s circumstances may disrupt long-standing habitual behaviours through habit discontinuity [26], thus leading to the development of new, potentially health-protective behaviours [27]. However, the extent to which individuals were taking the opportunity to exercise outdoors, or whether increased anxiety relating to fear of SARS-CoV-2 infection prevented this [20], are largely unknown.

Evidently, for many individuals, the closure of gyms, indoor athletic and leisure centres, as well as the cancellation of recreational sport, and restrictions on all but essential travel have likely caused a decline in the amount of physical activity they perform [28]. Indeed, it is possible that the amount of leisure time physical activity, particularly that which is of a vigorous intensity, alongside active transport and activities of daily living, have been greatly reduced [29]. The implications of this new, less active lifestyle are an increased risk of chronic disease [30]. Interestingly, this may also promote the suppression of immunological responses [31], further increasing the risk of contracting viral infections [32,33], such as COVID-19 [34].

Taking into account the aforementioned parameters, the COVID-19 lockdown/restriction measures undoubtedly had a major impact upon day-to-day life for a large proportion of the general population, with many individuals also experiencing increased childcare commitments [35] and a drastic shift in their work/life balance. Overall, this implies a reduction in active transport and activities of daily living, as well as reduced opportunities to participate in sport or exercise for leisure and health purposes. However, as aforementioned, the lockdown has also presented many individuals with increased availability of time, which could potentially lead to changes in their daily routines, including incorporating regular physical activity and exercise. Given this dichotomy of potential circumstances with regard to physical activity (i.e., an increase or decrease in relevant behaviours), and the effects that may persist beyond the end of the current pandemic (i.e., increased physical and psychological morbidity), it is important to precisely understand the trends in physical activity and sedentary behaviours as a result of the COVID-19 pandemic. As such, this systematic review aims to compare physical activity and sedentary behaviours in the general population during COVID-19-related lockdown, quarantine and self-isolation measures, to a time when no equivalent restrictions were in place. In the context of this systematic review, utilising broad inclusion criteria for included studies/participants will also allow for the inclusion of sub-analyses to investigate whether certain sub-populations (e.g., adolescents, older adults, and those with chronic disease) have increased or decreased their physical activity and sedentary behaviour to a greater or lesser extent due to the COVID-19-related lockdown/restrictions.

## 2. Materials and Methods

This systematic review was prospectively registered on the Prospero International Prospective Register of Systematic Reviews (CRD42021236563), and is reported according to the recommendations of the Preferred Reporting Items for Systematic Reviews and Meta-Analyses (PRISMA) statement [36].

### 2.1. Study Inclusion Criteria

An overview of the applied inclusion criteria for eligible studies is presented in Table 1. The objective of this systematic review is to assess the impact of COVID-19 lockdown measures on the amount of physical activity and sedentary behaviour performed by the general population. Therefore, a comparison will be made between time periods when lockdown measures were implemented against times when mandatory restrictions were not in place. Although the term “lockdown” is now synonymous with COVID-19, it appears to be an ill-defined construct and can encapsulate a wide range of restrictive measures [37]. For inclusion into this systematic review, a “lockdown” must include mandatory restrictive measures imposed by the corresponding government/authorities that are aimed at reducing the transmission of SARS-CoV-2, and these restrictive measures must be applied indiscriminately across the study population [38]. Conversely, for the purposes of this systematic review, a time when restrictive measures were not in place may be any time that “stay at home” orders were not implemented by the government of the country from which the participants were sampled. For inclusion in the systematic review, these criteria must be explicitly stated in the published article.

The primary outcomes of the present systematic review will be those relating to physical activity and sedentary behaviour (Table 1), and therefore studies must report either one or both to be eligible for inclusion. Physical activity is defined as any potential disruption to homeostasis by way of muscular contraction [39]; thus, this definition will encapsulate activities of daily living, active transport, planned exercise, and sport. By contrast, sedentary behaviour refers to waking behaviours/activities which demand an energy expenditure of 1.0–1.5 metabolic equivalent units (METs) whilst in a seated, reclining or lying posture [40]. Sedentary behaviour is often referred to as ‘sitting time’ and includes activities such as reading, driving a motor vehicle, watching television, computer usage, and other screen-based activities. Sedentary behaviour will be included, as opposed to physical inactivity (insufficient levels of physical activity to meet current levels of physical activity) [40,41].

Outcomes relating to physical activity and sedentary behaviours can be either device based or reported measures, with the former including measurements from accelerometers, pedometers, smart devices, global positioning system devices, or other physiological monitoring devices (e.g., heart rate monitors). Regarding reported measures, these include methods where data have been self-reported (e.g., questionnaires, activity logs, etc.), or obtained by proxy-reports [42].

### 2.2. Search Methods for Identification of Studies

The databases that will be searched are the Cochrane Central Register of Controlled Trials (CENTRAL) in the Cochrane Library, PubMed (via MEDLINE), EMBASE (via Web of Science), CINAHL and SPORTDiscus (via EBSCOhost), PSYCinfo, Coronavirus Research Database, Public Health Database, the Publicly Available Content Database (via ProQuest), SCOPUS, and Google Scholar. Grey literature will be searched using Bielefeld Academic Search Engine (BASE) and e-theses online service (EThOS) in the British Library. In addition, secondary searches of the reference lists of included/relevant papers will also be completed. As the first cases of COVID-19 were not reported until late December 2019 [43], and so as to ensure all related publications are captured, literature searches will only include papers published since November 2019 and up to the date of submission.

A search string was created for use in PubMed (Table 2), and then modified according to the syntax and appropriate headings of the other databases. Initial searches will be independently completed by two reviewers (CK and LL), and returned studies will be transferred into EndNote (Thompson Reuters, San Francisco, CA, USA) reference management software, where duplicates will be removed. Next, references will be imported into the Covidence (Veritas Health Innovation, Melbourne, Australia) systematic review management software, where further duplicates will be removed, and screening will be conducted; the screening of titles and abstracts and full-text screening will be completed independently by two reviewers. Disagreements with regard to study eligibility will be resolved by a discussion between reviewers, whilst unresolved disagreements will be arbitrated by a third reviewer. We will assess the level of agreement between the reviewers involved in study selection and those involved in the identification of the risk of bias, assessed using Cohen’s κ coefficient [44]. Given that a κ statistic score of 1 indicates perfect agreement and 0 equates agreement totally due to chance, we will categorise the scores as follows: poor (0), slight (0.1–0.2), fair (0.21–0.4), moderate (0.41–0.6), substantial (0.61–0.8), or near perfect (0.81–0.99).

### 2.3. Assessment of Risk of Bias in Included Studies

Observational studies are prone to distinct types of bias, namely selection and information bias [45]; it is therefore important to assess potential sources of bias in the included studies. To that aim, the Risk of Bias Assessment Tool for Nonrandomized Studies (RoBANS) [46] will be utilised. The RoBANS tool assesses six domains, namely the selection of participants, confounding variables, the measurement of exposure, the blinding of the outcome assessments, incomplete outcome data, and selective outcome reporting. Using RoBANS, two members of the review team will independently assess the risk of bias for each included study, grading each outcome within each study as having a “high“, “low”, or “unclear” risk of bias; any disagreements will be arbitrated by a third reviewer. The results of the risk of bias assessment will be presented in a “risk of bias” table within the final review.

### 2.4. Data Extraction and Analysis

Data extraction will be completed independently by two reviewers, and verified by an additional reviewer using a standardised form. Information extracted will include publication details, the characteristics of the study population (including country of study and time period(s) of data collection), study length, sample size analysed, behaviour examined (i.e., physical activity or sedentary behaviour), data collection methods (i.e., device-based or self-report), and the units of measurement (i.e., minutes per day/week, MET-minutes per day/week). Once all extracted data have been verified, a narrative synthesis will be completed, including summary tables. Depending on whether sufficient data are available, pooled effect estimates and their 95% confidence intervals (CIs) will be presented. The pooled effect estimates will be presented in a random effects meta-analysis, according to DerSimonian and Laird [47] (including Forest plots), using Review Manager (v5.4.1, Copenhagen: The Nordic Cochrane Centre). This method incorporates an assumption that each study is estimating different, yet related, effects; the standard errors of the study-specific estimates will be adjusted to incorporate a measure of the extent of variation, or heterogeneity, in the intervention effect from each individual study (tau-squared). Meta-analytical methods involving continuous outcomes (e.g., time in physical activity, energy expenditure or sedentary behaviour) assume that the data are normally distributed [48]. Data that are clearly skewed, or results that are reported with median and range values when non-parametric tests were used for analysis will be excluded from the meta-analysis.

For the assessment of physical activity, it is anticipated that, across the eligible studies, there will be multiple methods used to record physical activity. Accordingly, summary statistics for these data will be presented using the standardised mean difference (SMD) and Hedges’ *g* effect size [48]. However, should sufficient studies (≥2) utilise the same measurement methods, smaller analyses will be completed using the mean difference between comparators, whilst—where units of measurement can be transformed (e.g., minutes per day into hours per day)—values will be converted to reflect the most common measure before the meta-analysis is conducted.

### 2.5. Investigation of Heterogeneity

The presence of statistical heterogeneity will be determined using a chi-squared (χ^2^) test. Accordingly, a low *P* value or large χ^2^ value relative to its degrees of freedom provide confirmation of heterogeneity in the effects of the exposure that are not due to chance [48]. To describe the percentage of variability caused by heterogeneity rather than sampling errors, the *I*^2^ statistic [49] will be used. Heterogeneity will be interpreted according to the Cochrane Handbook recommendations as follows: 0–40% “might not be important”, 30–60% “may represent moderate heterogeneity”, 50–90% “may represent substantial heterogeneity”, and 75–90% “considerable heterogeneity” [48]. Should a large number of studies be included within the analyses, a degree of heterogeneity is likely to be present, and this may be further compounded by different methods of data collection and inter-population variability [50]. As such, should there be evidence of at least substantial heterogeneity, its source will be investigated through an analysis of the study population groups and the modality of data collection. In addition, a sensitivity analysis, which removes the largest outlier, will also be completed; if there is no substantial reduction in the degree of heterogeneity, it will also be investigated in subgroup analyses.

### 2.6. Subgroup and Sensitivity Analyses

In the context of this systematic review, where sufficient data are available (i.e., ≥2 studies), subgroup analyses will be performed for: participant age (<18 years, ≥18 years to <65 years, and ≥65 years), body mass index (BMI < 25 kg/m^2^, >25 kg/m^2^ to <30 kg/m^2^, and ≥30 kg/m^2^), data collection method (e.g., self-reported and device-based data), and any clinical populations (e.g., adults with pertinent cardio-metabolic diseases). Outcomes will be separated by subgroup and summary statistics will be presented for each relevant subgroup. Where insufficient data are available for a sub-analysis, findings will be reported qualitatively.

When statistical effects are identified within the primary analysis, sensitivity analyses will be conducted to test their robustness to different conditions. Accordingly, studies with a small sample size will be removed, as well as studies with at least one domain that is deemed at a high risk of bias. Where relevant, a further sensitivity analysis will be completed, which will only include data from studies that used validated self-report questionnaires, and studies with data from bespoke questionnaires will be excluded.

### 2.7. Assessment of Publication Bias

Publication bias will be investigated by visual inspection for asymmetry in funnel plots, with the degree of asymmetry assessed by Egger’s regression intercept test [51]. These investigations will only be completed should there be a sufficient number of studies for each outcome, that is, ≥10 studies will be required to detect an asymmetrical funnel [52].

## 3. Discussion

To the best of our knowledge, this PROSPERO-registered systematic review will be the first to review one complete year of published data on the impact of COVID-19-related movement restrictions on physical activity and sedentary behaviour. As such, our planned systematic review will provide a detailed overview of the relevant available literature one year after COVID-19 was declared a pandemic by the World Health Organisation (WHO) on 11/03/2020 [53]. The findings of our systematic review will provide up-to-date evidence to comprehensively inform the general public, clinical practitioners, public health agencies, researchers, policymakers and other stakeholders regarding the impact of lockdown/restriction measures upon physical activity and sedentary behaviour within the general population. Moreover, analyses of various populations (e.g., pertinent clinical conditions, age categories, or geographical regions) will allow for future public health interventions, which aim to promote physical activity and/or reduce sedentary behaviours, to be targeted at those who have been most affected.

The expected limitations of our planned systematic review are mainly dependent on the methods and quality of the published studies included. Indeed, eligible studies are likely to include multiple measurement tools/methods to capture physical activity and sedentary behaviours, which may make direct comparisons between studies difficult. Whilst it is expected that some of the included studies will have captured physical activity data using device-measured methods, it is more likely that several relevant studies utilised self-report methods, such as online questionnaires, which not only increase the reach and diversity of recruitment [54], but also reduce the need for physical contact with the research team, thereby lowering the risk of exposure to SARS-CoV-2 for study participants. However, there are certain limitations with self-report data, since physical activity is often over-reported [55], whereas sedentary behaviour is usually under-reported [50]. Moreover, recall bias, when participants are being asked to retrospectively recall their physical activity behaviours prior to COVID-19 lockdown measures, may also impact the findings of eligible studies with such self-reported outcomes. Finally, any attempted meta-analyses will depend the identification of a sufficient number of eligible studies that clearly define their participants and associated lockdown/restriction measures. Despite such unavoidable limitations, this systematic review is expected to constitute the most comprehensive and up-to-date synthesis of published evidence on the changes in physical activity and sedentary behaviour in the population due to the imposed COVID-19-related movement restrictions.

## 4. Conclusions

The COVID-19-related restriction measures had an abrupt and, in many cases, prolonged impact upon daily life for a large proportion of the general population in multiple countries worldwide. Overall, on the one hand, this acutely reduced opportunities to participate in sport or exercise for leisure and health purposes, but, on the other, has also provided many individuals with an increased availability of time to potentially engage in more regular physical activity and exercise. Given these opposing effects, understanding the exact impact of such mandated restriction measures on physical activity and sedentary behaviours is of great importance to better prepare for potential future waves of this or other pandemics. This planned systematic review will include relevant data from eligible studies covering more than a 12-month period since the start of the COVID-19 pandemic, and thus is expected to generate the most up-to-date findings in order to better characterise the impact of such lockdowns/restrictions on physical activity and sedentary behaviours.

## Figures and Tables

**Table 1 ijerph-18-05251-t001:** Eligibility criteria for inclusion in the present systematic review on the changes in physical activity and sedentary behaviour due to enforced coronavirus disease 2019 (COVID-19)-related lockdown/restriction measures.

Domain	Inclusion Criteria
Population	All individuals, regardless of age, medical status and geographic location are eligible for inclusion.
Exposure	COVID-19-related lockdown/restriction measures.
Comparator	Any period of time where COVID-19-related lockdown restriction measures were not being enforced.
Outcomes	Physical activity: measurement can include energy expenditure, metabolic equivalents, time performing an activity, and steps per day. Sedentary behaviour: measurement can include sitting time and summed time spent engaged in sedentary behaviours (e.g., screen time, passive transport).
Study design	Observational studies: cross-sectional studies, prospective cohort, and retrospective cohort studies.
Publication type	Published and unpublished grey literature are eligible for inclusion. No language restrictions will be imposed upon the search criteria, but only papers published in English will be included.

**Table 2 ijerph-18-05251-t002:** PubMed (via MEDLINE) search strategy.

(“COVID-19”[MeSH Terms] OR “Coronavirus”[MeSH Terms] OR “COVID*”[Title/Abstract] OR “Novel coronavirus”[Title/Abstract] OR “2019 novel coronavirus”[Title/Abstract] OR “2019-nCoV”[Title/Abstract] OR “SARS-CoV-2”[Title/Abstract] OR “Coronavirus disease 19”[Title/Abstract])
AND
(“Self-isolation”[Title/Abstract] OR “Isolation”[Title/Abstract] OR “Quarantine”[Title/Abstract] OR “Lock*”[Title/Abstract] OR “Restrict*”[Title/Abstract] OR “Measure*”[Title/Abstract] OR “Curfew”[Title/Abstract] OR “Confinement”[Title/Abstract])
AND
(“Exercise”[MeSH Terms] OR “Exercise movement techniques”[MeSH Terms] OR “Exercise Therapy”[MeSH Terms] OR “Exercise”[Title/Abstract] OR “Physical education and training”[MeSH Terms] OR “Physical fitness”[MeSH Terms] OR “Physical fitness”[Title/Abstract] OR “Physical exertion”[MeSH Terms] OR “Sport*”[MeSH Terms] OR “Sport*” [Title/Abstract] OR “Physical activit*”[Title/Abstract] OR “Walking”[MeSH Terms] OR “Walk*”[Title/Abstract] OR “Resistance Training”[MeSH Terms] OR “Muscle training”[Title/Abstract] OR “Strength training”[Title/Abstract] OR “Endurance training”[Title/Abstract] OR “Interval training”[Title/Abstract] OR “Intermittent training”[Title/Abstract] OR “Fitness”[Title/Abstract] OR “Swimming”[MeSH Terms] OR “Swim*”[Title/Abstract] OR “Bicycling”[MeSH Terms] OR “Bicycl*”[Title/Abstract] OR “Cycling”[Title/Abstract] OR “Cycle”[Title/Abstract] OR “Strengthening”[Title/Abstract] OR “Sedentary lifestyle”[Title/Abstract] OR “Sedentary”[Title/Abstract] OR “Physical* inactiv*”[Title/Abstract] OR “Sitting time”[Title/Abstract] OR “Sitting*”[Title/Abstract] OR “Home exercise”[Title/Abstract] OR “Home training”[Title/Abstract] OR “Yoga”[Title/Abstract])
AND
(“Observation* ”[Title/Abstract] OR “Cross-sectional”[Title/Abstract] OR “Cohort”[Title/Abstract])

## Data Availability

Not applicable.

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
