# Peer review of "Changes in Physical Activity and Sedentary Behaviour Due to Enforced COVID-19-Related Lockdown and Movement Restrictions: A Protocol for a Systematic Review and Meta-Analysis"

_ijerph, 2021, doi:10.3390/ijerph18105251_

Round 1

Reviewer 1 Report

Dear authors

The study topic is quite interesting and the introduction and methods are easy to read and to understand and follow international scientific rules.

I have one minor recommendation:

  • I would include the calculation of cohen´s kappa and intraclass correlation to evaluate the data codification/collection.

Kind regards

Author Response

We would like to sincerely thank the Reviewer for reviewing our work, and for these supportive comments and the helpful suggestion that allowed us to improve the submitted paper. 

As suggested, we have now added the following to the methods section (please also see lines 186-190 of the revised manuscript):

“We will assess the level of agreement between the reviewers involved in study selection and those involved in identification of risk of bias being assessed using κ Cohen’s coefficient [44]. Given that a κ statistic score of 1 indicates perfect agreement and 0 equates agreement totally due to chance, we will categorize the score as follows: poor (0), slight (0.1-0.2), fair (0.21-0.4), moderate (0.41-0.6), substantial (0.61-0.8), or near perfect (0.81-0.99).”

Reviewer 2 Report

This is well conceptualised and well-structured Systematic Review and Meta-Analysis protocol which should result in a worthwhile review. 

I have mainly very minor and only one more significant concern

Line 186 the table):  there is some redundancy. For example,  the search term “Physical Activit*” [Title/Abstract] will also capture “Physical Activity” [Title/Abstract] , or is that a typo and it should have been “Physical Activity” [MeSH Terms] ?  Likewise, in the pair “Sedentary lifestyle”[Title/Abstract] OR Sedentar*[Title/Abstract]  Sedentar* will capture both…

Also, check for presence of quotation marks throughout. This whole table needs a clean edit.

The only concern that I have is that by focussing on Title/Abstract or MeSH Terms, the authors will exclude studies /observations where the paper does  not focus on the changes in Physical Activity and Sedentary Behaviour, but reports on these in the context of a wider assessment. That can only be found through full text search (which Google Scholar allows).

Minutiae

Line 169 comma missing

Lines 327–331 stray text

Author Response

We would like to sincerely thank the Reviewer for reviewing our work, and for the provided comments and helpful suggestions that gave us the opportunity to revise and improve the submitted paper.

In response to these comments/suggestions, we have made the following revisions in Table 2:

1) We acknowledge the redundancy of the search term “Physical Activity” [Title/abstract], and so this has been now removed from the search string. We also tested replacing this with “Physical Activity” [MeSH Terms], but the search results were unaffected, hence the decision to remove it completely.

2) There was a typo identified for the search term Sedentar* [Title/Abstract]; this should not have an asterisk, but should have read “Sedentary”. We have amended this term in the search string and have tested this to ensure that the results differ to the “Sedentary lifestyle” search term.

3) As suggested, quotation marks have been added to each individual search term in Table 2 for consistency.

4) The PubMed search string that is provided in Table 2 has been piloted extensively, and we believe

it to offer a robust solution to the completion of the planned searches. In response to the provided comment, we combined [MeSH Terms] with All-Fields in an exploratory search, but the number of articles returned were unmanageable and based on initial screening a large proportion of items were completely irrelevant. It is also worth noting that searches will be completed across additional databases, and their search parameters will be followed. This further increases our confidence that important eligible papers will not be missed during our searches.

5) In line 169, we have added the missing comma.

6) We have deleted the stray text that was in lines 327–331, and we have added “Not applicable” to both the “Informed Consent” and “Data Availability” statements (please also see lines 334 and 335 of the revised manuscript).

Round 2

Reviewer 1 Report

Dear authors,

Thank you for addressing my comments.

Kind regards

Reviewer 2 Report

The authors are to be commended for addressing the issues identified